# Winter Sports Injuries in Elite Female Athletes: A Narrative Review

**DOI:** 10.3390/ijerph20105815

**Published:** 2023-05-13

**Authors:** Cristina Rotllan, Ginés Viscor

**Affiliations:** Secció de Fisiologia, Departament de Biologia Cel·lular, Fisiologia i Immunologia, Facultat de Biologia, Universitat de Barcelona, 08028 Barcelona, Spain

**Keywords:** alpine ski, snowboarding, ski jumping, cross-country ski, female, elite, injury incidence, injury risk

## Abstract

There is a lack of reviews covering the topic of the parallel high prevalence of injuries in female winter sport elite athletes. We aimed to review the data on incidence and patterns of injuries in female athletes participating in official competitions of winter sports. We conducted a comprehensive literature search on epidemiological data and etiological information on alpine skiing, snowboarding, ski jumping and cross-country skiing. The most common location of injury was the knee among skiers and ski jumpers and the incidence of severe ACL events was 7.6 per 100 ski racers per season (95% CI 6.6 to 8.9) in female alpine skiers. Snowboarders and cross-country skiers were more affected in the ankle and the foot. The most common cause was contact trauma with stagnant objects. The injury risk factors include training volume, knee pre-injuries, the period of the season and the technical equipment. Females are at greater risk of suffering from overuse injuries during the competitive season, as opposed to male athletes who are more likely to suffer from traumatic injuries. Our findings can be used to inform coaches and athletes and to guide future injury prevention plans.

## 1. Introduction

Physical activity has many health benefits, including a reduced risk of chronic disease development and the prevention of injuries [1]. Compared with the general population, musculoskeletal injuries in elite athletes can provoke a missed opportunity to train and compete, thus greatly influencing their professional careers [2]. The participation of females in professional winter sports has increased in the last decades at the Olympic Winter Games, World Cups and at National European and U.S. competitions [3,4,5,6,7]. Despite this considerable growth, the majority of the research regarding musculoskeletal injuries has been based on male athletes. Consequently, there is not enough injury data and knowledge about musculoskeletal injuries regarding in the female elite population. It has been reported that the injury incidence in winter sports among females is 2.10 injuries per 1000 athlete days [8]. Multiple elements can be the cause of these injuries including internal risk factors (i.e., sex, age, physical fitness) and external risk factors (i.e., ski bindings, telemark, weather) [3,4,7,9].

Alpine skiing and snowboarding are characterized by downhill descents at high speed, where the more prone anatomical regions to injuries are the knee, the back, and the hip [10]. Ski jumping discipline is characterized by jumps on hills of various lengths (85–109 m) in which speed can reach between 85 and 88 km/h. It has been supposed, not based on any scientific data, that women have more injury risk than men, however, the maximum distance for women is capped at 95% the distance of men’s [6]. The knee is the most common site of injury in women and therefore injury prevention is oriented to prevent jumps longer than hill size. Cross-country skiing is a Nordic skiing discipline [5], that is carried out on groomed trails by using light skis and boots and bindings which can be fixed or not [11]. The most injured anatomic locations are the ankle and foot, more commonly in women than men [12]. Elite athletes described the injury prevention process differently based on their sports experience and the injuries they suffered along their careers [13]. The first essential stages in preventing athletes’ injuries are screening, identifying, and correcting dangerous movement patterns [14]. For these reasons, it is key that injury prevention strategies must be tailored according to sex and sport modality, and they should be incorporated into warm-up routines [14]. The aims of this paper are (1) to delve deeper into the incidence, prevalence, type and principal causes of winter sports injuries in female elite athletes, which participated in official competitions and (2) to identify the main injury risk factors for the different winter sports.

## 2. Materials and Methods

This study followed the Scale for the Assessment of Narrative Review Article (SANRA) guidelines. Three electronic databases were searched for original and review articles: PubMed, SPORTDiscus and Scopus. We filtered for documents reporting information on elite winter sportswomen and for documents written in English and published in the last 5 years (2018–2022). We applied this time cut-off due to the scarcity of literature regarding injury characteristics in female elite athletes in previous years. Therefore, the following search strategy was used: (injuries OR injury risk) AND (skiing OR snowboarding) AND (female OR women) AND elite. The lists of the studies were also reviewed to ensure the content of relevant information. Inclusion criteria for the studies were: (1) epidemiological and etiological original and review articles, (2) documents adding information on the injury risk factors, (3) studies on winter sports disciplines related to skiing and snowboarding, and (4) papers discussing relevant injury information disaggregated by sex. The only exclusion criterion was content not relevant for this paper such as the treatment of injuries or papers focused only on preventive measures. The workflow for the literature search and selection of studies is presented in Figure 1.

Data extraction was conducted from every pertinent article. All articles included in this study described epidemiologic characteristics (incidence, prevalence, severity, type, cause, risk ratio of injury) and injury risks factors (sex, age, training load, competition season vs. off-season, weather, maturity status, pre-injuries, physical test, technical characteristics, training vs. competition).

Among all studies included just three were conducted exclusively on the female population, the rest reported on both sexes. The majority of the articles analyzed basic data separately by sex, however, there is limited information on injury mechanisms and risk factors in women skiers and snowboarders.

## 3. Results

After deleting the duplicates, the total number of studies included was n = 17 (original article n = 16, systematic review and meta-analysis n = 1), which described epidemiologic information and potentially associated factors. Moreover, we classified the reports by sport discipline including alpine skiing (n = 12), snowboarding (n = 4), ski jumping (n = 3) and cross-country skiing (n = 5).

### 3.1. Alpine Skiing

Alpine skiing is one of the most popular winter sports, although its practice exposes their athletes to a high risk of injury [3,4,5,9,10,15,16]. According to the recent results of the retrospective study on the Beijing 2022 Olympic Winter Games, alpine skiing was the second highest sport related to the number of injuries (n = 53). Therefore, in general, and regarding athletes suffering at least one injury, the percentage of sex-specific injuries was 13.3% of females and 9.5% of men. The Risk Ratio (RR) was 1.40, 95% CI: 1.14–1.72, *p* = 0.001. The incidence of injuries in females participating in alpine skiing was important (n = 24). Information related to injury location adds that the knee was the most usual anatomical location (n = 17) and, according to injury type, muscle and tendon strain (n = 17) were the most common kind of injuries, but data are not separated by sex [5]. Male and female skiers are included together in the majority of the epidemiological studies, although we found only one research exclusively based on elite female alpine skiers [15]. To our knowledge, during the last 5 years, only one systematic review and meta-analysis aimed to research the injury incidence in skiers, which examined sex-specific data, where the pooled incidence among females was 2.10 injuries per 1000 athletes/days [8].

From 1997 to 2020, Austrian Sky teams suffered an increase in the risk and the incidence of severe injury events [3,4]. The total incidence of injuries per 1000 runs, in competition, was 1.74 for female runners, with this being higher than the male incidence in all teams (Team Europa Cup, Team National Junior, Team World Cup). The female-male risk ratio (RR) was 1.36 [4]. Another interesting finding was the incidence of severe anterior cruciate ligament (ACL) events, with 7.6 per 100 ski racers/season (95% CI 6.6 to 8.9) among females. Specifically, Female World Cup alpine ski racers had a 1.65 times higher risk than their male counterparts to suffer ACL injuries [3]. However, in contrast to this finding, during the Olympic Winter Games in Pyeongchang 2018 [17,18], no differences were reported in overall injury incidence by sex in skiers, although significant differences in injury risk in females were found in other sports [18]. The injury incidence during the Paralympic Winter Games in Pyeongchang 2018 [17] was 23.1 in para-alpine skiing and the percentage of athletes with injury was 21.3%, although results were not analyzed separately by sex. Therefore, para-alpine skiing was in first position in female participation compared with other sports (n = 40).

In a prospective cohort study during five entire seasons (2013–2018) within the French National Alpine Ski Team B (European Cup), [10] there were no sex differences regarding the injury profile in terms of the circumstances, severity, locations and injured structures. However, the injury incidence per 100 athletes per season was comparable between men (121.5) and women (129.6), with the female-male RR being 1.07 and the female relative risk being 1.5 (0.93–2.42). Considering the period of the year, the incidence in the winter competitive season was 77.8 and thus higher than during the summer training season (51.9) for women skiers. These authors also found that the severe injury incidence was twice higher in winter vs. summer, which may be understood because, during the winter season, the number of severe knee injuries increased substantially, though data were not separately analyzed by sex (Table 1). Moreover, the gradual onset of injuries increased more in summer than in winter. A similar overall pattern of results was obtained in a prospective study, where traumatic injuries were more prone to appear during the competition period and overuse injuries during the off-season [15]. Interestingly, in female skiers, the results were the opposite, the incidence of severe traumatic injuries was higher during the off-period whereas overuse injuries were prevalent during the competition season. Another investigation in to entirely female elite alpine skiers [19], showed an average 2-weekly prevalence of overuse injuries during the off-season, with the knee being the most common location of the injury with 28.7% (95% CI, 24.9 to 32.5), followed by the back and the hip. Technically trained skiers were more frequently affected by back overuse injuries than skiers trained in speed disciplines. No significant differences at *p* < 0.05 were found according to the number, duration and severity of injuries [19]. Significant associations between total training volume and previous severe knee injuries with the occurrence of knee overuse complaints were also reported. Also, total training volume was found to be related to the cumulative severity score of knee overuse injuries as revealed after a Spearman’s rank correlation analysis (r = 0.536, *p* = 0,005, n = 26) [19].

The literature on the risk of injuries among youth ski racers is scarce. Research studying the relationship between biological maturity status, anthropometric characteristics and injury risk showed that maturity offset, changes in height and leg length were significantly different from injured than non-injured skiers. According to the fitness parameters, the only significant injury risk factor was changes in jump agility (*p* = 0.03) however, data distinguishing between men and women were not reported [9]. It should be noted that, although data were not disaggregated by sex, high training load (intensity and volume) did not show associations with increased injury risks in a prospective study of youth skiers, in which women reported ten traumatic and three overuse injuries, with the knee being the most affected anatomical location (39%) [16]. Some research performed during competition events like the Youth Olympic Games in Lausanne revealed that the female injury rate was 7.7% (in comparison to 3.9% in male athletes) and the incidents occurring in the Giant Slalom and Super giant slalom disciplines affected 78% of women racers and 62% of men racers. These results were not substantially different to previous Youth Olympic Games (Innsbruck 2012, Lillehammer 2016) nor in incidence nor regarding the effect of the sex of the participants [20].

### 3.2. Snowboarding

Snowboarding has progressively gained interest and the number of participations in international competitions is steadily increasing, such as during the Paralympic Winter Games in Pyeongchang 2018, where 14 females participated in this discipline. It is worth mentioning that para-snowboarding had the highest overall percentage of athletes with an injury (33.3%), and the majority of all these injuries (80%) affected lower limbs, head, neck and face [17].

In that same year, at the Olympic Winter Games [18] 1210 of the participants were women (42%). Adjusting for sport, there was no difference in overall injury incidence between women and men: RR = 1.15. Injury incidence proportions per 100 athletes in snowboard disciplines in total, not separately by sex, were snowboard cross (25.7), snowboard slopestyle (21.2), followed by snowboard halfpipe and snowboard big air and the last position was for snowboard slalom ranging from two to six injuries per 100 athletes. The most common injury mechanism in snowboard disciplines was contact with a stagnant object [18]. Additionally, other characteristics of a cause of injury were established during Pyeongchang 2018 Paralympic Winter Games as technical faults or difficulties, athlete being out of control and not knowing how the injury occurred [17]. In terms of the onset of injury in the snowboarding slalom, competition injuries were less common than training injuries (0 vs. 3) [18]. These results are consistent with a previous report, which described, in terms of the modality, but not of the sex, a higher injury proportion in snowboard cross and slopestyle [5]. In relation to the high-priority injuries during the most recent Olympic Winter Games in Beijing, snowboarding was the discipline with the highest rate of emergency medical services (n = 12), with 42.9 injuries per 100 athletes. The ankle was the anatomical part most affected in overall snowboarders and impingement was the most common type of injury. A recent systematic review on this issue provided an injury incidence of 3.99 per 1000 athletes and day in snowboarders in general and the most common cause of injury in these athletes was contact trauma [8].

### 3.3. Ski Jumping

Women’s ski jumping participation in the World Cup is relatively recent. Therefore, the literature shows only two prospective cohort studies applied to the female population during three seasons (2017–2018 to 2019–2020), although these provide valuable information [6,7]. A total of 54 injuries were produced, out of 205 athletes over these seasons. The accumulative injury incidence was 26.3 per 100 athletes per season. An important percentage of these injuries were acute (83.3%) and, regarding the injury location, the knee was the most common (n = 18, 33%), of which 10 of these were ACL ruptures [7]. These significant trends affecting this particular knee ligament could be explained because 77.8% of severe injuries involved this articulation. Severe injuries were defined as those injuries that did not allow the affected athletes to train and compete for more than 28 days [7]. Furthermore, equipment failure, adverse conditions on the in-run or out-run, and snowy/windy/cloudy weather were involved as relevant causes of injury events (Table 1). This study demonstrates also how the telemark landing and ski bindings may be related to knee injury [7].

### 3.4. Cross-Country Skiing

Relevant information was reported after a 52-week prospective study on 74 adolescent cross-country skiers and eight ski orienteers, all together considered endurance skiers (n = 82), of which 42 of the participants were women (Table 1) [21]. The injury incidence rate for endurance skiers was 2.7 per 1000 h of practice in the sport, where injury prevalence was 21.4% and the most common new injury location was affecting the foot, when not separately analyzed by sex. Moreover, endurance skiing was the sport with the longest median time to first injury (41 weeks) as compared with the shortest, 9 weeks, in handball players, also not separately by sex [21]. Overall, these findings agree with those further reported by Worth et al. (2019) who studied the injury incidence and risk factors in elite North American cross-country skiers: ankle and foot accounted for 42.6% of all lower limb injuries in men, while 90.5% of all new injuries reported by women affected the lower extremity [12]. Another 17-week prospective cohort study in a population of cross-country skiers showed a 9% of average weekly prevalence of injuries in females. Concerning age, athletes in development were less likely to suffer substantial and overuse injuries than senior athletes in general, and regarding sex differences, females were more likely to incur substantial health problems [22]. Reports from the last two Olympic Winter Games, 2018 and 2022, described an injury incidence ranging from 2 to 6 per 100 athletes [18], with five female injuries occurring in this sport [5]. No information on injury location and type separately by sex was reported [5,18]. The time of injury onset during the Pyeongchang 2018 Olympic Winter Games showed a higher proportion of injuries during training (70.6%) compared with the competition (23.5%) [18].

## 4. Discussion

We conducted a narrative review aimed at comparing the incidence and main risk factors and associated circumstances of winter sports injuries in female elite athletes. Due to limited research focused on women, there is a knowledge gap regarding injury mechanisms in female skiers. The pooled incidence of injuries among female winter athletes was 2.10 per 1000 athletes/days [8], with the lower extremity being the most frequently affected body segment in skiers and snowboarders [3,7,12,16,19]. Specifically, the knee was the most commonly affected location for alpine skiers and ski jumpers, moreover, ACL injury was the most frequent affectation. Austrian female alpine skiers during World Cup had a higher risk ratio (1.65) for ACL injuries compared to males [3], therefore during World Cup ski jumping competitions, ACL ruptures were an important proportion of injuries [7]. Remarkably, past severe traumatic knee injuries affected 57.7% of female alpine participants, with 30.8% of them undergoing knee surgery, including ACL reconstruction [19]. In line with this trend, future prevention plans should prioritize addressing the risk of the first severe knee injury and monitoring those skiers with a history of that injury due to their elevated risk of re-injury.

Cross-country skiers showed a higher propensity for injuries in the ankle and foot [12], while ankle injuries were the most common among snowboarders [5]. Considering the varying locations of injuries in different sports, it is crucial to consider the technical characteristics and biomechanics of each activity to understand the injury mechanism, particularly for ACL injuries, and develop strategies to decrease the injury rate. Additionally, the influence of sex as a factor in the incidence of injuries among female athletes should be considered. In the last years, the hypothesis that female hormones can affect the risk of injuries on ligaments, muscles or tendons has been proposed. However, it is important to note that there is a lack of high-quality methodology in the majority of available studies [23]. The Austrian Ski Federation conducted a 22-season injury surveillance study to determine the incidence and sex-specific risk ratio of acute severe injury. They found that the total incidence of injuries per 1000 female runs was 1.74, which was higher than that of male alpine skiers [4]. Despite the implementation of instructional programs on ski technique and approach, course and jump design, snow preparation and specific athletic conditioning programs, the average seasonal incidence of acute injuries in Austrian alpine ski racers of the World and European Cup has increased from 11 to 23 [3]. Moreover, training load has been associated with knee overuse complaints in female alpine skiers during the off-season [19]. Thus, injury prevention efforts should be tailored and designed to consider the sex of the participants and season period-specific aspects [15]. Training load monitoring could also be used as a preventive measure and the frequency, duration and intensity of the training load could be increased to improve the performance [8]. It has been demonstrated that jumping, running and flexibility exercises, as well as balance and strength training, minimize the likelihood of injuries, such as ACL damage. One of the first steps should be to identify and correct dangerous movement patterns like the dynamic valgus. A possible option for future studies could be the use of online resources for appropriate, comprehensive preventative strategies freely available and tailored to the various sports disciplines [14]. Additionally, more attempts should be made to gain a deeper understanding of ACL injury mechanisms and their long-term effects on injured athletes, in order to improve clinical practice rehabilitation and readaptation.

In this study, we have observed that the risk of injuries among alpine skiers varies throughout the season, and it is influenced by whether the athletes are in the competition period or preparation period [15]. Females are more likely to be affected by overuse injuries during the competition season and traumatic injuries during the off-season. This could be due to differences in the training load, with the summer season characterized by high-intensity physical activity, with many hours on snow and occasional competitions, while the competition season involves lower training intensities and more physical work preservation [24]. For that reason, it is necessary to consider season periods and sex-specific aspects to design new prevention strategies. Another crucial factor affecting injury risk is poor snow conditions towards the end of the competition season, which can increase the likelihood of injuries [16]. It must be noted that the weather contributed to the higher percentage of injuries, with 92% of the injuries in World Cup ski jumping competitions occurring in snowy, windy or cloudy conditions [7]. Although we are not able to control the weather, it is important to take it into account and postpone competitions if necessary.

We observed that delayed maturity status, shorter size and leg length are factors associated with increased injury risk in alpine skiers [9]. Conversely, higher age is a feature related to overuse injuries in cross-country skiing [22]. Additionally, we found that the jump agility test could be used as a predictor of injuries in youth ski racers, as proper timing of muscle actions and inter- and intra-muscular coordination of the relevant muscles is essential in injury prevention strategies. Incorporating neuromuscular training and agility test batteries into the development plans of youth ski racers is also recommended [9]. Furthermore, the American Medical Society for Sports Medicine recommends that youth athletes should not train more hours per week than their age [25].

Contrasting the main cause of injury between sports, we found that crash landings and ski bindings were the primary injury mechanisms in the sport of ski jumping, with most injuries occurring on the ski jump hill [6,7]. Future research should be aimed at understanding in more detail how to reduce these risks and design effective strategies dealing with technical and equipment failure and analyzing biomechanics during the landing phase. The principal cause of injury across sports studied was contact trauma with an inanimate object [18]. Therefore, it is essential to consider this aspect and focus injury reduction efforts on such circumstances.

This study is not without limitations. Precise data on female athletes were not available in many of the studies, and for this reason, some of the results were not presented here segregated by sex, mostly according to the cause and risk factors. Moreover, the reported incidence data were not always consistently presented in the same units by different sources, which made it difficult to compare studies and draw firm conclusions. Future studies should address these limitations by focusing on defining the etiology of winter sports injuries and elucidating injury mechanisms, particularly in the biomechanical analysis of knee injuries, especially ACL injuries. Furthermore, preventive injury strategies should be implemented separately for each sport, sex and age group to effectively reduce the incidence of injuries among female athletes.

## 5. Conclusions

To our knowledge, this is the first review on female musculoskeletal injuries in winter sports, evaluating the lower extremity as the most affected body part. Female skiers and ski jumpers suffered knee injuries as the most frequent event, with an incidence of severe ACL injuries at 7.6. per 100 ski racers per season. However, snowboarding and cross-country skiing had an even higher prevalence of ankle and foot injuries. The principal overall cause of injuries was contact trauma. Although, it must be mentioned that telemark landings and ski bindings were involved with knee injury mechanisms. Preventive measures should be focused on reducing the risk of female injuries by considering such factors as training load, neuromuscular conditions, previous knee injuries, the period of the year and technical issues.

## Figures and Tables

**Figure 1 ijerph-20-05815-f001:**
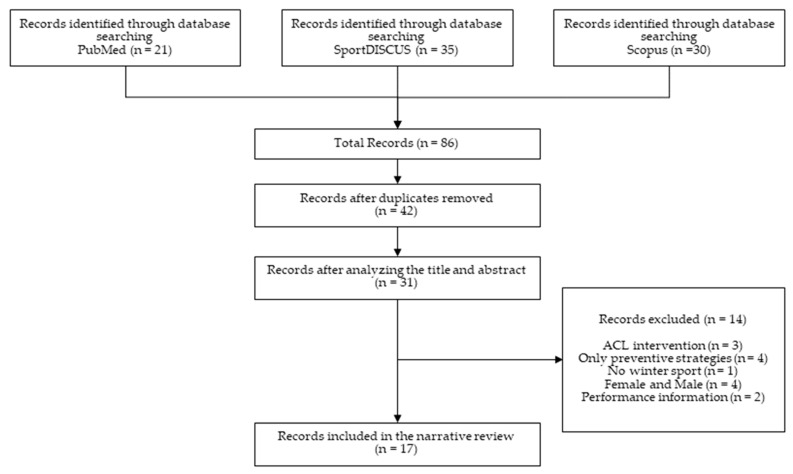
Flow diagram for search, identification and inclusion of literature cited.

**Table 1 ijerph-20-05815-t001:** Main injured causes and injured risk factors in different winter sports.

Winter Sports	Injury Causes and Circumstances (Female and Male)	Injury Risk Factors (Classified by Sex)
Alpine skiing	Contact with stagnant object [18] Competition & Training [5,18]	Female Sex [3,4,10] Higher training load (F) [19] Knee pre-injuries (F) [19] Traumatic during the off-season (F) [15] Overuse during competitive season (F) [15] Shorter size and leg length (F&M) [9] Worse jump agility (F&M) [9] Delay maturity status (F&M) [9]
Snowboarding	Contact with stagnant object [18] Athlete out of control [17] Technique faults and difficulties [17] Emergency medical services [5] Competition & Training [5,18]	Unknown
Ski jumping	Contact with stagnant object [18] Crash landings [7] Adverse conditions in-run/out-run [7] Bad weather [7] Competition & Training [5,18]	Unfavorable weather (F) [6,7] Equipment failure (binding) (F) [6,7]
Cross-country skiing	Contact with stagnant object [18] Competition & Training [5,18]	Pre-injuries (F&M) [12]

F, female; M, male.

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
