# Peer review of "Winter Sports Injuries in Elite Female Athletes: A Narrative Review"

_ijerph, 2023, doi:10.3390/ijerph20105815_

Round 1

Reviewer 1 Report

The authors address a gap in current knowledge and the premise for this study is therefore sound. There are some problems, however, with the aim, methods and presentation of the results.

My main concern with this study is the unclear aims. A review should have a clearly articulated aim, similar to any other research study. In the abstract, the authors state: “we aimed to review the data of incidence and patterns of injuries in female athletes practicing winter sports.” In the introduction, the authors write: “… aims of this paper are 1) to delve deeper on the incidence, prevalence, type and principal causes of snow sports injuries in females and 2) to identify the main injury risk factors for the different winter sports.” The aim in the introduction is still not a well-articulated research question. Is this limited to female professional athletes? Any limitation re: country or time period? Will injury patterns in females be compared with men? There are many irregularities in definitions, starting with interchangeable use of ‘winter sports’, ‘snow sports’, and the actual key words used in the search, which only included snowboarding and skiing. The word ‘elite’ is similarly unclear and the search did not include any other key words for professional athletes. There should be a clearly articulated definition/scope around the study group and sports, and a comprehensive search strategy to match.

Not every review needs to be a systematic review, but a more clearly defined aim, a structured approach and a more comprehensive search strategy are required for addressing this topic.

Author Response

Many thanks for your review and advice. To improve and clarify the aims of the study we redefined it in the introduction: we specified the population as female elite athletes, not limiting the country and period. We considered elite athletes who are dedicated professionally to sport and they participated in national teams and competed in national cups or international competitions like World’s Cup and Olympic Winter Games. Moreover, to enclose the research question we have decided to use the term ‘winter sports’. 

The search was limited to female athletes, although considering the limited studies performed only in females (3 studies from 17) and the limited data analyzed separately by sex we showed some relevant results comparing females and men.

We initially decided to limit the search to snowboarding and skiing, taking into account the large quantity of winter sports disciplines, which are the most popular winter sports. Once, we read all the studies we considered also relevant to include studies on ski jumping because they are addressed exclusively to elite women in World Cup. Furthermore, we decided to include reports on cross-country skiing because new injury locations were analyzed in females and, in addition, more information was compared between both sexes.

Considering your review and those from other reviewers, and also to enhance the narrative review, we decided to include a brief description of every sport (characteristics and biomechanics).

We hope with these changes the manuscript has improved their comprehension and structure.

Reviewer 2 Report

The paper is very well-written in term of the language and organization. However, the scientific merit is of concern. The current manuscript is a narrative review on the winter sports, the incidence rate is summarized from previous studies. However, the figures were not well documented and tabulated. In addition, the content is not providing enough new information to the incidence rates which were reported by FIS or other organizations. I am sorry that I cannot agree to the paper at the current stage.

Author Response

Many thanks for your review and your suggestions for the improvement of our document. According to your comments and leading to enhance the quality of the manuscript we provide additional information. As suggested, we added the cites of each finding in the table and we unified the incidence rates when it has been possible, to make results more comprehensible.

We really appreciate your suggestions and those from other reviewers. We hope, with these changes the quality and the comprehension of the manuscript have improved and you agree to accept our narrative review.  

Reviewer 3 Report

Overall this was a very successful manuscript.

Regularly the authors interchange gender and sex. I imagine, based on regulatory issues, this paper is about sex and not gender or gender identity. Change throughout.

There are several incidents where the syntax is a problem, impacting meaning. For example, line 27 should be "in the last decade" or "over the lat few decades." Or line 31, "regarding gender factor should be "regarding gender."

Abstract formatting does not require subheadings or numeration. Consider removing.

Methods - line 44 revise to "Primarily, we used a filter of 5 years prior to the period of publication (2018-2022), female sex, and English language."

Where the articles appraised using a systematic approach? If so, this should be included in the methods.

Consider a visual depiction of the search strategy and inclusion/exclusion process for article selection.

Is Table 1 from another source. Each finding should be cited in the table.

There seem to be several terms used throughout to depict exposures and incidence. These should be more clearly communicated. For instance, a reader may try to draw a false equivalency to # of injuries per skier to per hours of exposure. 

The Discussion becomes choppy and lacks cohesion through the lat three paragraphs. Suggest finding a way to integrate lines 251-253 and 255-257 into the other sections and advancing the content in the paragraph about limitations (lines 259-263).

Author Response

Overall this was a very successful manuscript. Regularly the authors interchange gender and sex. I imagine, based on regulatory issues, this paper is about sex and not gender or gender identity. Change throughout.

Many thanks for your recommendation, you are right. We changed it as you suggested.

There are several incidents where the syntax is a problem, impacting meaning. For example, line 27 should be "in the last decade" or "over the last few decades." Or line 31, "regarding gender factor should be "regarding gender."

We apologize for this error, and we have corrected the text as suggested.

Abstract formatting does not require subheadings or numeration. Consider removing.

We changed the abstract to a non-structured format.

Methods - line 44 revise to "Primarily, we used a filter of 5 years prior to the period of publication (2018-2022), female sex, and English language."

We checked and corrected this line, we removed “Primarily”.

Where the articles appraised using a systematic approach? If so, this should be included in the methods.

Many thanks for your comment. We consider our article a narrative review because it addresses a topic in wider ways and is more descriptive than a systematic one, which answers a narrow question and uses a more abundant literature search.

Consider a visual depiction of the search strategy and inclusion/exclusion process for article selection.

Many thanks for your recommendation, we added a visual depiction for the search strategy (Figure 1).

Is Table 1 from another source? Each finding should be cited in the table.

We appreciate your comment and we agree with that, as also one of the rest of the reviewers, therefore we added the corresponding literature sources in the table.

There seem to be several terms used throughout to depict exposures and incidence. These should be more clearly communicated. For instance, a reader may try to draw a false equivalency to # of injuries per skier per hour of exposure. 

We agree with you. However, the different sources did not allow to express the risk and incidence in equivalent terms. We specify this point in the study limitations the difficulties for the reader and this could be improved in future research.

The Discussion becomes choppy and lacks cohesion through the last three paragraphs. Suggest finding a way to integrate lines 251-253 and 255-257 into the other sections and advancing the content in the paragraph about limitations (lines 259-263).

We appreciate your view and we modified and integrated these paragraphs in another part of the discussion to achieve more cohesion, while we tried to improve the paragraph declaring the limitations of this study.

Reviewer 4 Report

General comments

The work presented by the authors is well written and covers an interesting and relevant topic. 

Specific comments

Abstract: line 8: the authors may consider to add a brief sentence underlying the reason/importance of their work (e.g., lack of reviews covering the topic with a parallel high prevalence of injuries in female winter sport athletes). Line 15: the authors may consider to use "injury risk factors" rather than injury risks and eventually training volume instead of training hours. Also, "season" is not clear as an injury risk factors.

Introduction: The authors may consider to implement their introduction covering aspects as the following:

Consider to include a brief description/introduction: considering the characteristics of winter sport disciplines (e.g., biomechanics) what are the main anatomical structures at higher injury risk and why/how. Maybe considering what has been revealed from male studies.

Consider to extend your background regarding impacts of injuries on career and injury prevention/rehabilitation strategies.

Methods: methods section is clearly described.

Results: statement in lines 77-78 is not clear. The authors describe the most common injurym anatomical location (i.e., the knee) but it is also mentioned about muscle and tendon strain injuries that more describe type of injury. consider to divide in two sentences and better specify 

Discussion:

Consider to change training hours with training volume.

Lines 232-237: consider to extend the discussion regarding mechanisms leading to injuries and eventually proposing options to characterize these mechanisms in future studies.

Lines 241-242: Better utilization of terms describing training, training volume, training intensity, training load describe different concepts. Therefore, the sentence "volume of training load" may be incorrect conceptually.

Author Response

The work presented by the authors is well written and covers an interesting and relevant topic. 

Specific comments

Abstract: line 8: the authors may consider to add a brief sentence underlying the reason/importance of their work (e.g., lack of reviews covering the topic with a parallel high prevalence of injuries in female winter sport athletes). Line 15: the authors may consider to use "injury risk factors" rather than injury risks and eventually training volume instead of training hours. Also, "season" is not clear as an injury risk factor.

We agree with that and we considered the changes in line 8 and 15. We changed “season” to “period of the season” to better specify the concept, which means the injury risk change according to the period of the season (summer vs winter).

Introduction: The authors may consider to implement their introduction covering aspects as the following:

Consider to include a brief description/introduction: considering the characteristics of winter sport disciplines (e.g., biomechanics) what are the main anatomical structures at higher injury risk and why/how. Maybe considering what has been revealed from male studies.

Consider to extend your background regarding impacts of injuries on career and injury prevention/rehabilitation strategies.

Thank you for your suggestions, we added more information according to your suggestion about the characteristics of sports disciplines and strategies for injury prevention.

Methods: methods section is clearly described.

Results: statement in lines 77-78 is not clear. The authors describe the most common injury anatomical location (i.e., the knee) but it is also mentioned muscle and tendon strain injuries that more describe type of injury. consider to divide in two sentences and better specify 

Many thanks for your amendment, we specified this in a better way.

Discussion:

Consider to change training hours with training volume.

We agree with that and we changed the terms throughout.

Lines 232-237: consider to extend the discussion regarding mechanisms leading to injuries and eventually proposing options to characterize these mechanisms in future studies.

Thank you so much for the comments we added more information regarding possible prevention strategies and future plans of action.

Lines 241-242: Better utilization of terms describing training, training volume, training intensity, training load describe different concepts. Therefore, the sentence "volume of training load" may be incorrect conceptually.

We apologize for this error. We have corrected the text as suggested.

Round 2

Reviewer 1 Report

The revised manuscript is improved but editing to improve the grammar is still required. Please also revise the title: ".. injuries in female elite population..." is confusing. The study if focused on a population of elite athletes not females elites. 

Author Response

Thank you again for your help, and especially for your warning regarding the title of the article. We have reviewed again the text with the help of an English native colleague. Several grammatical editions appear in red (all other previous marks on R1 have been deleted for clarity)

Reviewer 2 Report

The manuscript has been greatly improved after the revision. Although the novelty of study is still of concern, I agree for publication if EiC had accessed the suitability of scope. Thank you.

Author Response

Thank you for your valuable contribution to the improvement of our article. We hope that after your help and addressing the comments and suggestions of the rest of the reviewers, the article will be suitable for publication in this present form.